# Exploratory Pilot Study on the Serum Ceramide (16:0) to Sphingosine-1-Phosphate Ratio as a Potential Indicator of Lupus Nephritis and Disease Activity

**DOI:** 10.3390/ijms262411957

**Published:** 2025-12-11

**Authors:** Ji-Won Kim, Seung Hyun Kim, Wook-Young Baek, Ju-Yang Jung, Hyoun-Ah Kim, Chang-Hee Suh

**Affiliations:** 1Department of Rheumatology, Ajou University School of Medicine, 164 Worldcup-Ro, Yeongtong-Gu, Suwon 16499, Republic of Korea; jwk722@naver.com (J.-W.K.); arikato83@naver.com (W.-Y.B.); serinne20@hanmail.net (J.-Y.J.); nakhada@naver.com (H.-A.K.); 2Translational Research Laboratory for Inflammatory Disease, Clinical Trial Center, Ajou University Medical Center, Suwon 16499, Republic of Korea; kimsh@ajou.ac.kr; 3Department of Molecular Science and Technology, Ajou University, Suwon 16499, Republic of Korea

**Keywords:** systemic lupus erythematosus, lupus nephritis, sphingolipids, biomarker, disease activity

## Abstract

Sphingolipids are essential for cellular structure and signaling, and recent evidence implicates them in chronic inflammation. We hypothesized that altered sphingolipid metabolism contributes to the disease activity of systemic lupus erythematosus (SLE). Serum sphingolipids were quantified by liquid chromatography–tandem mass spectrometry in 38 female SLE patients (11 with lupus nephritis [LN], 27 without LN) and 30 age-matched healthy controls (HCs). Serum ceramide (Cer)16/sphingosine-1-phosphate (S1P) ratios were elevated in SLE compared to HCs (0.33 [0.26–0.38] vs. 0.25 [0.21–0.3], *p* = 0.019). Notably, Cer16/S1P levels were significantly higher in the LN group (0.33 [0.26–0.38]) than in non-LN SLE (0.27 [0.2–0.34], *p* = 0.027). ROC analysis showed good diagnostic potential for LN (AUC = 0.739). Cer16/S1P correlated positively with disease activity markers, including erythrocyte sedimentation rate (r = 0.519, *p* = 0.001), SLE Disease Activity Index 2000 (SLEDAI-2k) score (r = 0.547, *p* < 0.001), anti-double stranded DNA antibody levels (r = 0.359, *p* = 0.027), and the Systemic Lupus International Collaborating Clinics/American College of Rheumatology Damage Index (r = 0.327, *p* = 0.045). The serum Cer16/S1P ratio may serve as a surrogate marker of disease activity in patients with LN.

## 1. Introduction

Systemic lupus erythematosus (SLE) is a chronic autoimmune disease, characterized by the production of autoantibodies and widespread inflammation that affects multiple organ systems. SLE predominantly occurs in women of reproductive age, and exhibits a broad spectrum of clinical manifestations ranging from mild cutaneous lesions to severe multiorgan involvement, including nephritis, arthritis, and neuropsychiatric complications [1]. The heterogeneity of clinical manifestations and fluctuating disease activity present substantial challenges in understanding the underlying molecular mechanisms and identifying robust biomarkers [2].

Sphingolipids, a class of bioactive lipids, serve as essential signaling molecules that regulate diverse biological and structural functions within cells. As integral components of cell membranes, they contribute to cellular integrity and mediate key processes such as apoptosis, proliferation, and immune regulation [3]. Ceramides (Cer) are well documented for their role in promoting pro-inflammatory responses and inducing apoptosis, whereas sphingosine-1-phosphate (S1P) plays a critical role in lymphocyte trafficking, vascular homeostasis, and immune modulation [4]. Given their roles in immune regulation, dysregulation of sphingolipid metabolism may contribute to various pathological conditions, including cancer, neurodegenerative disorders, metabolic diseases, and, more recently, chronic inflammatory diseases including SLE [5,6].

Emerging evidence suggests that sphingolipid imbalance may also have clinical relevance in SLE, particularly in the context of lupus nephritis (LN), the most severe manifestation of the disease [7,8,9]. However, the diagnostic and prognostic value of sphingolipid metabolites in human SLE—including their association with disease activity indices and organ involvement—remains incompletely elucidated. Moreover, sphingolipid levels can be influenced by various factors including diet, medication, and genetic variation, and their specific contributions to disease pathogenesis may differ across ethnic and geographic populations [10,11].

Based on previous studies highlighting the role of metabolomic and immune biomarkers, including sphingolipids [12,13], this study was designed as an exploratory pilot investigation to evaluate the feasibility of sphingolipid profiling in Korean patients with SLE. We hypothesized that altered sphingolipid metabolism is associated with disease activity indices and organ involvement in Korean patients with SLE, particularly LN, and aimed to evaluate preliminary data that may support future large-scale prospective studies.

## 2. Results

### 2.1. Baseline Characteristics and Clinical Manifestations of Patients with SLE and Healthy Controls

The study population comprised 38 patients diagnosed with SLE, including 27 without LN, 11 with biopsy-confirmed LN, and 30 age-matched healthy controls. Table 1 summarizes the baseline characteristics and clinical manifestations of patients with SLE and healthy controls. No significant differences were observed in age, sex distribution, or disease duration between the SLE subgroups or when comparing each subgroup with the healthy control group. Lipid profile analysis revealed a trend toward higher total and LDL cholesterol levels in the control group, although these differences were not statistically significant (169.0 ± 26.4, 168.2 ± 21.3, and 190.5 ± 42.0 mg/dL for SLE without LN, SLE with LN, and healthy controls, respectively; *p* = 0.063). Clinically, no significant differences were observed in the prevalence of mucocutaneous symptoms, arthritis, serositis, or hematological involvement between SLE patients with and without LN. Laboratory findings indicated higher mean anti-double-stranded DNA (anti-dsDNA) levels in the LN group than those in the non-LN group (32.8 IU/mL vs. 13.1 IU/mL; *p* = 0.052). As expected, the proportion of patients with proteinuria >0.5 g/day was significantly higher in the LN group than that in the non-LN group (54.5% vs. 11.1%; *p* = 0.004). Although the Systemic Lupus Erythematosus Disease Activity Index 2000 (SLEDAI-2k) and Systemic Lupus International Collaborating Clinics Damage Index (SDI) scores tended to be higher in the LN group, these differences were not statistically significant (SLEDAI-2k: 5.70 ± 4.09 vs. 8.18 ± 4.42, *p* = 0.107; SDI: 0.26 ± 0.81 vs. 0.64 ± 1.29, *p* = 0.383 for SLE without LN vs. SLE with LN).

Patients with LN were more frequently prescribed immunosuppressive agents, particularly mycophenolate mofetil (63.6% vs. 0%; *p* < 0.001). The proportion of patients receiving glucocorticoids and their cumulative glucocorticoid doses tended to be higher in the LN group, although the differences were not statistically significant. In contrast, nonsteroidal anti-inflammatory drugs were more commonly used in the non-LN group (51.9% vs. 18.2%; *p* = 0.057). The percentages of patients taking lipid-lowering agents were 14.8% in the non-LN group, 18.2% in the LN group, and 6.7% in the healthy control group, with no statistically significant differences among the three groups (*p* = 0.222).

### 2.2. Sphingolipid Profile in Patients with SLE with and Without LN and Healthy Controls

The levels of S1P, sphingosine, ceramides, sphingomyelin (SM), and specific ceramide-to-S1P ratios were analyzed, and the results are presented in Table 2. No significant differences were observed in S1P, sphingosine, or ceramide species between patients with SLE with LN, patients without LN, and healthy controls. Similarly, most SM levels did not differ significantly between groups. However, when comparing all patients with SLE with healthy controls, SM (d18:0/24:0) levels were significantly lower in patients with SLE (16,719 [11,262–19,284] vs. 17,110 [9907–19,871], *p* = 0.027). Regarding ceramide-to-S1P ratios, Cer14/S1P (0.01 [0–0.02] vs. 0.01 [0–0.02], *p* = 0.031), Cer16/S1P (0.33 [0.26–0.38] vs. 0.25 [0.21–0.3], *p* = 0.019), and Cer24:1/S1P (1.26 [1.01–1.53] vs. 1.04 [0.8–1.31], *p* = 0.031) ratios tended to be elevated in the SLE with LN group compared to those in the healthy control group. Notably, these differences were not statistically significant when comparing SLE patients with and without LN, except for those with Cer16/S1P. The Cer16/S1P ratio was significantly higher in the SLE with LN group compared to both the SLE without LN group and the healthy control group (0.33 vs. 0.27 vs. 0.25, *p* = 0.027 and *p* = 0.019, respectively).

### 2.3. Diagnostic Value of Cer16/S1P Ratio in Identifying Lupus Nephritis

The diagnostic potential of Cer16/S1P, which exhibited the most pronounced difference between patients with SLE and healthy controls and showed significant variation within the SLE subgroup based on LN status, was assessed. The results are shown in Figure 1. Figure 1A presents a dot plot comparing Cer16/S1P values between patients with LN and healthy controls. Figure 1B shows the receiver operating characteristic (ROC) curve for serum Cer16/S1P in distinguishing patients with SLE with LN from healthy controls, demonstrating good diagnostic capability with an area under the curve (AUC) of 0.739 (95% confidence intervals (CI): 0.581–0.898). Given the greater clinical relevance of distinguishing lupus nephritis from non-renal SLE, we additionally performed ROC analysis comparing patients with LN and those without LN (Figure 1B). Cer16/S1P yielded an AUC of 0.736, indicating moderate discriminative ability, although interpretation should remain exploratory due to the limited sample size and wide confidence intervals (CI: 0.580–0.891). Collectively, these results indicate that Cer16/S1P may be the most promising serum sphingolipid biomarker for identifying LN among patients with SLE.

### 2.4. Correlation Between the Cer16/S1P Ratio and Clinical Features and Disease Activities in SLE

The correlations between Cer16/S1P and various clinical parameters in patients with SLE were analyzed, and the results are shown in Table 3. The Cer16/S1P demonstrated a statistically significant positive correlation with disease duration (r = 0.373, *p* = 0.021), anti-dsDNA levels (r = 0.359, *p* = 0.027), erythrocyte sedimentation rate (ESR) (r = 0.519, *p* = 0.001), creatinine (r = 0.373, *p* = 0.021), cumulative glucocorticoid dose (r = 0.467, *p* = 0.003), SLEDAI-2k scores (r = 0.547, *p* < 0.001), and Systemic Lupus International Collaborating Clinics (SLICC) damage index (SDI) (r = 0.327, *p* = 0.045). Conversely, a significant negative correlation was observed with hemoglobin (r = −0.419, *p* = 0.009). Notably, as the SLEDAI-2k score increased, reflecting higher disease activity, serum Cer16/S1P levels also increased, as illustrated in Figure 2A. To further explore this relationship, we compared the serum Cer16/S1P ratio between patients with moderate and higher disease activity, and those with low disease activity. Figure 2B displays that in patients with a SLEDAI-2k score of ≥6, the serum Cer16/S1P was significantly higher compared to those with low or no disease activity (0.34 vs. 0.22; *p* < 0.001).

## 3. Discussion

This study investigated the alterations in sphingolipid metabolism in SLE, focusing on the Cer16/S1P ratio and its association with clinical features. Our findings reveal several key insights into the potential role of sphingolipid dysregulation in SLE pathogenesis.

Lipid metabolism in autoimmune diseases, such as SLE, has received increasing attention over the past decade, particularly regarding hyperlipidemia and elevated cardiovascular risk due to disrupted lipid pathways; however, the role of sphingolipid metabolism remains relatively underexplored [14]. Given that genetic, environmental, and dietary factors influencing sphingolipid metabolism can vary across populations, investigating these markers in specific ethnic groups is crucial to refine their clinical applicability [15]. To our knowledge, this is the first study to examine sphingolipid profiles in Korean patients with SLE, highlighting the importance of ethnicity-specific research in validating and advancing lipid-based biomarkers for clinical use.

Firstly, we observed significant differences in the serum Cer16/S1P ratio between healthy controls and patients with SLE, especially among those with LN. These findings align with those of previous studies that reported altered sphingolipid profiles in patients with SLE, particularly in non-Korean populations [16]. Notably, most studies have observed a trend toward increased ceramide biosynthesis or enhanced S1P degradation, reflecting disruptions in sphingolipid metabolism associated with inflammation and immune dysregulation.

Secondly, the Cer16/S1P ratio demonstrated significant correlations with multiple clinical features in SLE patients. The positive correlation with SLEDAI-2k and SDI suggests that altered sphingolipid metabolism is associated with increased disease activity and cumulative damage in SLE. Furthermore, the correlation with ESR, Creatinine, and anti-dsDNA indicates a potential link between sphingolipid dysregulation and inflammation, renal involvement, and autoantibody production, respectively. The significant correlation with cumulative glucocorticoid dose may reflect the influence of treatment on sphingolipid metabolism or, conversely, the impact of the Cer16/S1P ratio on treatment response.

These findings suggest several potential mechanisms through which altered sphingolipid metabolism may contribute to the pathophysiology of SLE. Ceramides are bioactive lipids involved in various cellular processes, including inflammation, apoptosis, and immune cell signaling. Elevated ceramide levels can induce pro-inflammatory cytokine production and promote immune cell activation, potentially exacerbating the inflammatory and autoimmune responses in SLE. For example, studies have shown that ceramides activate the NLRP3 inflammasome, leading to caspase-1 cleavage and subsequent secretion of pro-inflammatory cytokines, a pathway closely linked to SLE pathogenesis [17]. In addition, oxidative stress, which is heightened in SLE, stimulates sphingomyelinase activity and enhances ceramide generation. Excessive ceramide accumulation, in turn, impairs mitochondrial function and promotes further ROS production, creating a vicious cycle that amplifies immune dysregulation and tissue injury [18]. Altered sphingolipid metabolism may also contribute to renal dysfunction. Sphingolipids are critical for maintaining the integrity of the glomerular filtration barrier, and disruption of this barrier can lead to proteinuria and kidney injury [19]. Notably, Cer24:0 has been identified as a potential marker of renal damage, supporting its role in LN pathogenesis [20]. Along with Cer16:0 and Cer18:0, Cer24:0 is among the ceramide species most frequently elevated in SLE, consistent with established roles in promoting inflammation, apoptosis, and tissue damage [21]. Similar findings have been reported in studies conducted in European countries and China, further emphasizing the potential universality of these metabolic alterations in SLE pathogenesis [18,20,21].

Accumulating evidence suggests that dysregulated sphingolipid metabolism plays a role in SLE. In Chinese patients with SLE, the composition of lipid species, including diacyl phosphatidylethanolamine, Cer22:0, and Cer24:1, was significantly altered [22,23]. In another study, lactosylceramide levels were found to have predictive value, and sphingolipidomics provided additional benefit over currently available tools for early diagnosis and prognosis in African American SLE patients with cardiovascular disease [24]. Similar dysregulation of ceramide metabolism has also been reported in other autoimmune diseases, including rheumatoid arthritis, Sjögren’s syndrome, systemic sclerosis, and systemic vasculitis, highlighting a potentially shared pathogenic role of ceramides in systemic autoimmunity [25,26,27,28,29]. Although several studies have reported alterations in ceramide and S1P levels in SLE and other autoimmune diseases [25,26], the direction and magnitude of these changes can vary depending on the specific disease and patient population. Even within the same disease, differences across ethnicities and cohorts have been observed, making it challenging to generalize a consistent pattern of sphingolipid dysregulation across all autoimmune conditions.

Surprisingly, a study conducted in Sweden reported that Cer16/S1P was the most effective discriminator between SLE patients and healthy controls, which is consistent with the findings of our study [9]. Traditionally, very long chain ceramides such as Cer24:0 have been considered more pivotal in driving inflammation, as supported by numerous studies [30]. However, recent findings, including ours, increasingly emphasize the significant role of Cer16:0 in the development of metabolic and autoimmune diseases, as well as its association with mortality in these conditions [31,32,33]. One possible explanation for this shift in focus is the prevalence of palmitate, the fatty acid component of Cer16:0, in Western diets [34]. As the Korean diet becomes westernized, the contribution of Cer16:0 to metabolic dysfunction and inflammation is likely to increase. Although direct mechanistic investigation was beyond the scope of this study, emerging experimental data indicate that Cer16:0 accumulation may contribute to renal injury through mitochondrial dysfunction, podocyte apoptosis, and activation of inflammatory pathways. Reports demonstrating increased Cer16 levels and CerS6 expression in kidney tissue and podocytes under metabolic or inflammatory stress further support a potential role of Cer16 in renal involvement, which may partly explain its stronger association with LN in our study [35]. Our findings likely reflect this trend, highlighting not only the impact of dietary and metabolic factors on the development of chronic diseases but also the potential significance of these factors as therapeutic targets.

Several studies have reported alterations in sphingolipid profiles following lupus treatment, supporting their potential use as biomarkers and therapeutic targets. Checa et al. reported significant increases in S1P levels and decreases in the ratios of Cer16:0 and Cer24:1 to S1P following treatment, highlighting the treatment-induced shifts in sphingolipid metabolism [9]. Similarly, a study investigating the effects of rituximab found that sphingolipids such as C16:0 dihydroceramide and C16:0 glucosylceramide were reduced, which was associated with disease improvement [36]. These findings highlight the dynamic nature of sphingolipid regulation in response to treatment and further emphasize its relevance in understanding the pathogenesis and progression of SLE. However, targeting sphingolipids for treatment is challenging because of their complex metabolism. The alteration of one sphingolipid may affect others, potentially leading to unintended side effects [37]. Additionally, sphingolipid metabolism can vary across different ethnic groups owing to genetic and environmental factors that may affect disease outcomes and treatment responses [38]. The complexity and diversity of sphingolipids, combined with challenges such as poor solubility and difficulty delivering many mimetics, have hindered their establishment as definitive therapeutic agents [39,40]. Nevertheless, sphingolipids remain a field with high potential for ongoing, careful, and rigorous research and development.

The clinical relevance of our study lies in the potential of the Cer16/S1P ratio as an exploratory biomarkers for disease activity in SLE. Although ROC analysis suggests that the Cer16/S1P ratio may have diagnostic potential for lupus, this finding should be considered preliminary and requires validation in larger, prospectively designed cohorts. Monitoring the Cer16/S1P ratio could potentially contribute to the assessment of disease activity or therapeutic response; however, this interpretation remains hypothesis-generating. The observed association with SLEDAI-2K scores highlights Cer16/S1P as a marker of moderate-to-high disease activity, which is consistent with findings from other studies. Given the current efforts to identify non-invasive predictive biomarkers for LN, our observation that Cer16/S1P was associated with renal involvement may provide a biological signal that warrants further investigation [41,42]. In addition, the Cer16/S1P ratio was modestly higher in patients with LN compared to those without LN, supporting its possible relevance to renal manifestations. However, due to the small sample size and cross-sectional design, this finding should be interpreted with caution and regarded as hypothesis-generating. Although applicability at the time of diagnosis could not be fully assessed owing to heterogeneous sampling timing, the biological plausibility observed suggests that Cer16/S1P may hold potential not only for monitoring disease flare or worsening activity but also as an early diagnostic indicator. Prospective longitudinal studies involving treatment-naïve patients and standardized sampling at diagnosis will be required to confirm this hypothesis.

The correlation between the Cer16/S1P ratio and disease duration is particularly noteworthy as it suggests that prolonged disease may be associated with cumulative alterations in sphingolipid metabolism, potentially driven by chronic inflammation, persistent immune dysregulation, or long-term treatment effects. The positive correlation between cumulative glucocorticoid dose and Cer16/S1P further raises the possibility that treatment influences sphingolipid biosynthesis pathways, indirectly reflecting the therapeutic burden and long-term impact of the disease. However, this also introduced a limitation to our study, as corticosteroids are known to affect lipid profiles. This limitation is particularly relevant when considering Cer16/S1P as a diagnostic biomarker. Ideally, sphingolipid profiles obtained at the time of diagnosis would provide a more accurate representation; however, because of the study design, many samples were collected after treatment initiation, making it challenging to fully exclude the effects of corticosteroids and immunosuppressants.

This study has several strengths, including comprehensive analysis of clinical features and the correlation of sphingolipid metabolites with various disease parameters. However, it also has some limitations. First, the sample size was relatively small, particularly among patients with LN, which may limit the generalizability of the findings and contribute to statistical instability. In line with this, although an exploratory multivariable analysis was attempted to adjust for potential confounders, model instability due to the small number of LN cases and quasi-complete separation resulted in unreliable estimates; therefore, the analysis was not included in the final results to avoid overinterpretation. This limitation was further compounded by the nature of the metabolomic analysis, as variations in sample collection timing can introduce significant variability, making it difficult to integrate newly acquired samples with previously collected data. Second, cross-sectional study design provides only a snapshot of sphingolipid levels at specific time points, preventing assessment of longitudinal changes and precluding causal inferences regarding the relationship between sphingolipid alterations and disease activity. Future prospective longitudinal studies with repeated sphingolipid measurements and follow-up of renal outcomes will be necessary to determine whether Cer16/S1P predicts flare or disease progression and to support outcome-based analyses. Third, potential treatment-related confounding cannot be fully excluded. Although we minimized the inclusion of patients on lipid-lowering agents and observed no significant differences between groups, complete exclusion was not feasible, and other medications or comorbidities, such as metabolic syndrome or cardiovascular disease, may have influenced sphingolipid metabolism.

Despite these limitations, a key strength of our study is its focus on the Korean population, which addresses a critical gap in the literature. Although sphingolipid alterations have been extensively studied in Western populations, they remain poorly understood in Asian cohorts, where genetic and lifestyle factors may uniquely affect lipid metabolism and immune responses. Future studies involve longitudinal cohorts of newly diagnosed treatment-naïve LN patients to assess dynamic changes in Cer16/S1P over time. Integration of multi-omics approaches, including metabolomics, transcriptomics, and proteomics, may provide a more comprehensive understanding of sphingolipid dysregulation. Mechanistic validation studies are also warranted to elucidate the causal role of Cer16/S1P in LN pathogenesis.

## 4. Materials and Methods

### 4.1. Participant Details and Clinical Evaluations

This study enrolled female patients aged 18 years and older who were diagnosed with SLE according to the 2012 SLICC criteria or the 2019 European League Against Rheumatism/American College of Rheumatology (EULAR/ACR) criteria and had been regularly followed at Ajou University Hospital (Suwon, Republic of Korea). Patients receiving a daily prednisolone-equivalent dose of >20 mg, those who were pregnant or breastfeeding, and those with recent infections, surgery, or other acute or significant comorbid conditions were excluded. Among the included patients, 11 had biopsy-confirmed LN. LN was defined based on renal biopsy findings compatible with the International Society of Nephrology/Renal Pathology Society 2018 classification.

The sample size was determined based on our available equipment, the experimental conditions for sphingolipid measurements, and the study period. Due to limited funding and the defined study duration, only samples collected within the past two years were included. During this period, 40 patient samples could be secured, and the experimental process allowed for an additional 30 samples; therefore, 30 age-matched healthy female controls from the same region were included to complete the sample set.

The flowchart is presented in Figure 3. The study protocol was reviewed and approved by the Institutional Review Board of Ajou University Hospital (IRB No. AJOUIRB-SMP-2020-294; approved on 26 August 2020). All procedures involving human participants were conducted in accordance with the ethical standards of the institutional and national research committee and with the 1964 Declaration of Helsinki and its later amendments. Written informed consent was obtained from all participants prior to enrollment. Clinical assessments were conducted during outpatient visits, to evaluate manifestations such as oral ulcers, rashes, arthritis, alopecia, and fever. Laboratory tests included complete blood count, ESR, renal function tests, urine protein-to-creatinine ratio, and complement levels (C3 and C4). Autoantibodies, including antinuclear antibodies (ANA) and anti-dsDNA, were measured. Disease activity and cumulative organ damage were assessed using SLEDAI-2K and SDI, respectively, based on clinical and laboratory data. Patients with SLEDAI-2K scores ≥6 were classified as having active disease.

### 4.2. Sample Preparation and Sphingolipid Analysis

Blood samples for sphingolipid analysis were collected prospectively after obtaining informed consent during routine morning laboratory testing, following an overnight fast of ≥8 h to minimize dietary influence on lipid levels. No biobanked or previously stored samples were used. All specimens were freshly processed, with serum separated by centrifugation at 4 °C and stored at −80 °C until analysis. Sphingolipids, including S1P and sphingosine, were quantified using the Mass Hunter Quantitative Analysis B.07.00 system (Agilent Technologies, Santa Clara, CA, USA), with D-erythro-Sphingosine C-17 as the internal standard (Cayman Chemical, Ann Arbor, MI, USA). Calibration curves were established through linear regression (r ≥ 0.99) over a concentration range of 0.1–1000 nM to ensure accurate quantification. Ceramides and SMs were analyzed using liquid chromatography-tandem mass spectrometry with a 1290 HPLC system (Agilent, Waldbronn, Germany) coupled to a QTRAP 5500 mass spectrometer (AB Sciex, Toronto, ON, Canada). The analysis was performed in the positive ion mode using multiple reaction monitoring. Data extraction and quantification were conducted using Analyst 1.5.2 software.

### 4.3. Statistical Analysis

Continuous variables were tested for normality using the Shapiro–Wilk test. Given the limited sample size and the non-normal distribution observed in several key variables, continuous data—including sphingolipid levels—were summarized as median with interquartile range and compared using the Mann–Whitney U test for two-group comparisons or Kruskal–Wallis test for comparisons among the three groups (SLE with LN, SLE without LN, and healthy controls). Categorical variables were expressed as counts and percentages and analyzed using the chi-square test or Fisher’s exact test, as appropriate. To assess the diagnostic potential of sphingolipids, ROC curve analysis was conducted, and the AUC was calculated. Spearman’s rank correlation coefficient was used to examine the association between sphingolipid levels and clinical parameters, including anti-dsDNA antibody levels, complement levels, ESR, and SLEDAI-2K scores. A preliminary exploratory multivariable logistic regression analysis including age and sex was attempted to evaluate potential confounding effects; however, model instability due to the limited number of LN cases and quasi-complete separation prevented reliable estimation. Therefore, these results were not included in the main analysis to avoid overinterpretation. Statistical significance was set at *p* < 0.05. All analyses were performed using SPSS software version 25.0 (IBM Corporation, Armonk, NY, USA).

## 5. Conclusions

In conclusion, our study highlights the potential of Cer16/S1P as a biomarker of LN and disease activity in patients with SLE from Gyeonggi Province, Korea. Among the various sphingolipid markers analyzed, Cer16/S1P was the most significantly elevated in patients with SLE with LN compared to those without LN and healthy controls, demonstrating good diagnostic performance in distinguishing LN from healthy individuals. Additionally, Cer16/S1P positively correlated with disease activity markers, including SLEDAI-2K, SLICC damage index, disease duration, anti-dsDNA, and ESR, while exhibiting a negative correlation with hemoglobin levels. We demonstrated that Cer16/S1P levels were significantly associated with disease activity indices. This suggests that Cer16/S1P may serve as a potential biomarker of disease activity in SLE, particularly in relation to renal involvement, although validation in larger longitudinal cohorts is needed.

## Figures and Tables

**Figure 1 ijms-26-11957-f001:**
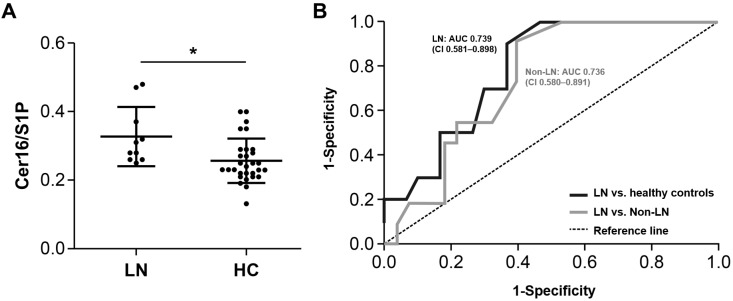
Elevated serum Cer16/S1P and its diagnostic performance for lupus nephritis. (**A**) Comparisons between SLE with LN and healthy controls. (**B**) Receiver operating characteristic curve for the serum Cer16/S1P. The solid line represents comparison between LN patients and healthy controls (AUC = 0.739, 95% CI: 0.581–0.898). The dashed line represents comparison between LN and non-LN SLE patients (AUC = 0.736, 95% CI: 0.580–0.891). Cer: ceramide, S1P: sphingosine-1-phosphate, SLE: systemic lupus erythematosus, LN: lupus nephritis, HC: healthy controls. * *p* < 0.05. Black circles represent individual patient values.

**Figure 2 ijms-26-11957-f002:**
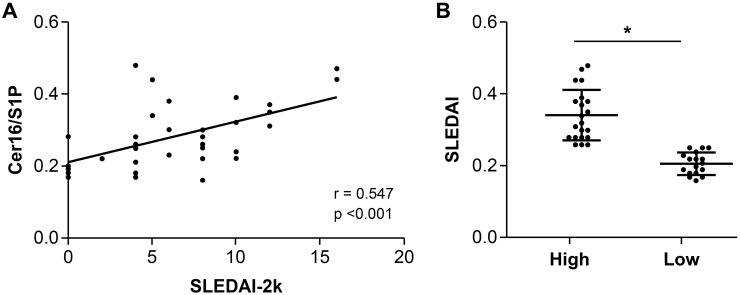
Correlation between serum Cer16/S1P and SLEDAI-2k scores. (**A**) Serum Cer16/S1P positively correlates with SLEDAI-2k scores. (**B**) Higher serum Cer16/S1P in SLE patients with SLEDAI-2k ≥ 6. Cer: ceramide, S1P: sphingosine-1-phosphate, SLEDAI-2k: Systemic Lupus Erythematosus Disease Activity Index 2000, SLE: systemic lupus erythematosus. * *p* < 0.05.

**Figure 3 ijms-26-11957-f003:**
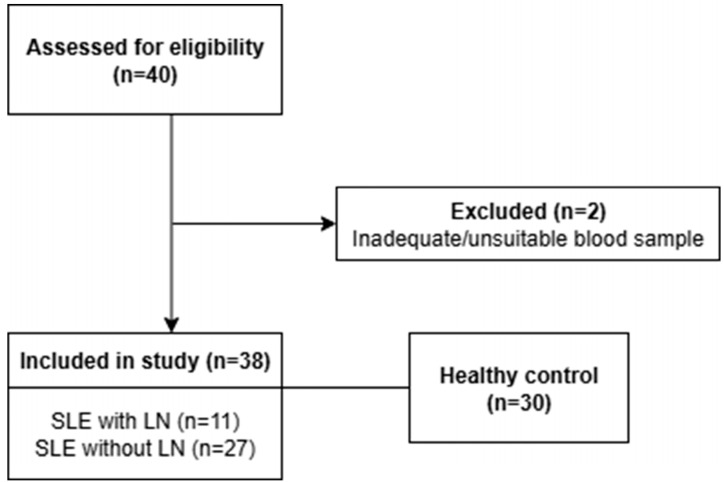
Flow chart of patient selection. SLE: systemic lupus erythematosus, LN: lupus nephritis.

**Table 1 ijms-26-11957-t001:** Summary of demographic and clinical data for patients with SLE with and without nephritis, and healthy controls.

Variable	SLE Without LN (N = 27)	SLE with LN (N = 11)	Healthy Controls (N = 30)	*p*-Value
Age, years	41 (25–57)	47 (34–58)	46 (30–55)	0.380
Female, no. (%)	24 (88.9)	10 (90.9)	28 (93.3)	0.305
Disease duration, months	122 (67–177)	90 (83–163)	NA	0.479
Lipid profile				
Total cholesterol, mg/dL	171 (149–190)	164 (154–205)	180 (164–203)	0.063
LDL cholesterol, mg/dL	104 (67–119)	97 (91–132)	95 (80–121)	0.665
HDL cholesterol, mg/dL	62 (56– 67)	53 (45–60)	66 (56–78)	0.423
Triglyceride, mg/dL	70 (43–93)	55 (38–90)	73 (59–103)	0.100
Clinical manifestations				
Mucocutaneous, no. (%)	20 (74.1)	9 (81.8)	NA	0.611
Arthritis, no. (%)	18 (66.7)	9 (81.8)	NA	0.350
Serositis, no. (%)	2 (7.4)	1 (9.1)	NA	>0.999
Hematologic, no. (%)	10 (37.0)	7 (63.6)	NA	0.135
Central nervous system, no. (%)	0 (0)	1 (9.1)	NA	0.289
Laboratory finding				
Leukocyte/μL	4403 (3378–5060)	3910 (3550–4950)	NA	0.116
Lymphocyte/μL	1478 (1322–1872)	1517 (1129–1810)	NA	0.857
Hemoglobin, g/dL	13.4 (12.9–13.9)	12.4 (11.1–12.9)	NA	0.054
Platelets, ×10^3^/μL	245 (197–263)	211 (200–236)	NA	0.319
ESR, mm/h	8 (2–14)	16 (8–20)	NA	0.461
ANA positivity, no. (%)	24 (88.9)	10 (90.9)	NA	>0.999
Anti-dsDNA (IU/mL)	13.1 ± 24.4	32.8 ± 32.8	NA	0.091
Anti-dsDNA Ab positivity, no. (%)	8 (29.6)	7 (63.6)	NA	0.052
Anti-Sm Ab positivity, no. (%)	3 (11.1)	1 (9.1)	NA	>0.999
APL positivity, no. (%)	5 (18.5)	3 (27.3)	NA	0.548
Low complements (C3 < 90 mg/dL or C4 < 10 mg/dL), no. (%)	10 (37.0)	7 (63.6)	NA	0.135
Proteinuria (mg/day)	0.2 (0.1–0.5)	0.8 (0.3–1.4)	NA	0.204
Proteinuria >0.5 g/day, no. (%)	3 (11.1)	6 (54.5)	NA	**0.004**
SLEDAI-2k	5 (4–8)	8 (4–12)	NA	0.107
SDI	0 (0–1)	0 (0–1)	NA	0.383
Treatment				
NSAID, no. (%)	14 (51.9)	2 (18.2)	NA	0.057
Glucocorticoids, no. (%)	19 (70.4)	9 (81.8)	NA	0.467
Cumulative GC dose, g, (prednisolone-equivalent)	2.0 (0–3.0)	4.0 (1.5–10.5)	NA	0.168
Hydroxychloroquine, no. (%)	23 (85.2)	10 (90.9)	NA	>0.999
Azathioprine, no. (%)	3 (11.1)	1 (9.1)	NA	>0.999
Mycophenolate mofetil, no. (%)	0 (0)	7 (63.6)	NA	**<0.001**
Calcineurin inhibitor no. (%)	3 (11.1)	3 (27.3)	NA	0.329
Methotrexate no. (%)	2 (7.4)	0 (0)	NA	>0.999
Lipid lowering agents	4 (14.8)	2 (18.2)	2 (6.7)	0.222

SLE: systemic lupus erythematosus, LN: lupus nephritis, LDL: low-density lipoprotein, HDL: high-density lipoprotein, ESR: erythrocyte sedimentation rate, ANA: anti-nuclear antibody, dsDNA: double-strand deoxyribonucleic acid, Ab: antibody, Sm: Smith, APL: antiphospholipid antibodies, C3: complement 3, C4: complement 4, SLEDAI-2k: SLE disease activity index 2000, SDI: systemic lupus international collaborating clinics damage index, NSAID: nonsteroidal anti-inflammatory drugs, GC: glucocorticoid, NA: not applicable. Continuous variables are presented as median (interquartile range), and categorical variables as number (percentage). Bold values indicate *p* < 0.05.

**Table 2 ijms-26-11957-t002:** A detailed table listing sphingolipid species and their concentrations in patients with SLE with and without nephritis, and healthy controls.

Sphingolipids (ng/mL)	SLE Without LN (N = 27)	SLE with LN (N = 11)	HCs (N = 30)	*p*-Value ^1^	*p*-Value ^2^
S1P	702 (650–770)	645 (590–710)	705 (660–760)	0.208	0.380
Sphingosine	3.6 (2.2–5.5)	2.8 (2.0–3.6)	3.4 (2.2–4.3)	0.155	0.680
Cer14:0	6.8 (5.5–9.0)	7.7 (6.5–9.2)	7.0 (5.6–8.2)	0.512	0.671
Cer16:0	182 (160–210)	199 (175–225)	178 (150–205)	0.372	0.140
Cer18:0	98 (73–122)	104 (91–120)	93 (69–118)	0.591	0.081
Cer20:0	111 (89–138)	118 (105–146)	118 (100–138)	0.754	0.319
Cer24:0	2056 (1691–2764)	1999 (1585–2720)	2191 (1918–3015)	0.313	0.145
Cer24:1	694 (599–952)	803 (618–1010)	774 (616–999)	0.406	0.387
SM (d18:0/16:0)	26,208 (19,683–33,211)	27,281 (19,534–35,017)	29,922 (19,189–37,893)	0.474	0.583
SM (d18:0/18:0)	5413 (4687–7063)	6161 (5659–7514)	6674 (5589–8058)	0.231	0.140
SM (d18:0/24:0)	16,719 (11,262–19,284)	17,110 (9907–19,871)	19,853 (12,648–20,134)	**0.027**	0.053
SM (d18:0/24:1)	35,097 (26,109–41,893)	34,560 (30,205–37,874)	34,134 (28,648–38,216)	0.986	0.934
Cer14/S1P	0.01 (0–0.02)	0.01 (0–0.02)	0.01 (0–0.02)	0.110	**0.031**
Cer16/S1P	0.27 (0.2–0.34)	0.33 (0.26–0.38)	0.25 (0.21–0.3)	**0.027**	**0.019**
Cer18/S1P	0.14 (0.12–0.16)	0.16 (0.14–0.19)	0.13 (0.11–0.15)	0.189	0.107
Cer20/S1P	0.17 (0.15–0.18)	0.2 (0.17–0.22)	0.17 (0.16–0.18)	0.215	0.283
Cer24/S1P	3.08 (2.79–3.34)	3.11 (2.83–3.4)	3.27 (2.89–3.72)	0.691	0.198
Cer24:1/S1P	1.09 (0.99–1.25)	1.26 (1.01–1.53)	1.04 (0.8–1.31)	0.111	**0.031**

SLE: systemic lupus erythematosus, LN: lupus nephritis, HC: healthy control, S1P: sphingosine-1-phosphate, Cer: ceramide, SM: sphingomyelin. Data are presented as median (interquartile range). *p*-value ^1^ indicates comparison between SLE without LN vs. Healthy controls and *p*-value ^2^ indicates comparison between SLE with LN vs. HCs. Bold values indicate *p* < 0.05.

**Table 3 ijms-26-11957-t003:** Correlation between sphingolipid and SLE disease activity parameters.

Parameter	Cer16/S1P
	Correlation Coefficient, r (*p*-Value)
Age	−0.026 (0.836)
Disease duration	**0.373 (0.021)**
Anti-dsDNA Ab	**0.359 (0.027)**
Complement 3	−0.121 (0.470)
Complement 4	0.009 (0.956)
Leukocyte	−0.057 (0.736)
Lymphocyte	0.016 (0.923)
Hemoglobin	−0.419 (0.009)
Platelet	−0.043 (0.796)
ESR	**0.519 (0.001)**
Creatinine	**0.373 (0.021)**
Cumulative glucocorticoid dose	**0.467 (0.003)**
SLEDAI-2k	**0.547 (<0.001)**
SLICC damage index	**0.327 (0.045)**

Cer: ceramide, S1P: sphingosine-1-phosphate, dsDNA: double-strand deoxyribonucleic acid, Ab: antibody, ESR: erythrocyte sedimentation rate, SLEDAI-2k: SLE disease activity index 2000. Bold values indicate *p* < 0.05.

## Data Availability

The original contributions presented in this study are included in the article. Further inquiries can be directed to the corresponding author.

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
