# Peer review of "Exploratory Pilot Study on the Serum Ceramide (16:0) to Sphingosine-1-Phosphate Ratio as a Potential Indicator of Lupus Nephritis and Disease Activity"

_ijms, 2025, doi:10.3390/ijms262411957_

Round 1
Reviewer 1 Report
Comments and Suggestions for Authors
This study explores the role of sphingolipids and Cer16/S1P ratio as a potential indicator of lupus nephritis (LN) and disease activity in systemic lupus erythematosus (SLE). The topic is relevant and the biochemical rationale is interesting. However, several methodological and interpretational limitations substantially weaken the strength of the conclusions. The findings appear preliminary and largely observational, and the clinical applicability of the Cer16/S1P ratio remains unclear.
Major Comments
- Small sample size and lack of statistical adjustments
The most significant limitation is the small sample size, which makes it difficult to draw robust conclusions. All reported associations are unadjusted, and potential confounders were not accounted for. As a result, it is difficult to determine whether Cer16/S1P provides information beyond established clinical and laboratory parameters.
- Unclear clinical added value
Although the Cer16/S1P ratio was significantly higher in SLE patients with LN compared to those without LN and healthy controls, the authors do not assess its incremental diagnostic value. Does Cer16/S1P outperform or add to existing biomarkers such as the SLEDAI score, complement levels (C3), or anti–dsDNA antibodies? How does it compare with established LN diagnostic tools, such as proteinuria, urinary sediment, or biopsy findings? Without comparative or multivariable analyses, the clinical utility of this marker cannot be determined.
- Timing in disease course is unclear
Patients were included at an average of 9 years after SLE diagnosis, which complicates interpretation. At what stage of SLE or LN progression is Cer16/S1P informative? Is this marker useful early in the disease, at flare, or only in long-standing SLE? Clarification is crucial for understanding how this biomarker could be used clinically.
- No assessment of association with clinical outcomes
The study does not evaluate whether Cer16/S1P correlates with meaningful clinical outcomes such as kidney function, renal flare/response, progression to ESKD. Without outcome associations, the interpretation of Cer16/S1P as an “indicator” of LN or disease activity is limited.
Minor Comments
- The title states that the Cer16/S1P ratio is a “potential indicator of lupus nephritis,” but this only becomes clear after reading the manuscript. Consider revising for clarity and accuracy.
- Given the small cohort, the statistical analysis should rely on non-parametric methods, and continuous variables should be presented as median and interquartile range (IQR) rather than means. This would more accurately reflect the distribution of data and improve interpretability.
Reviewer 2 Report
Comments and Suggestions for Authors
The manuscript investigates the Cer16:0/S1P ratio as a biomarker for lupus nephritis (LN) and SLE disease activity. While the topic is clinically relevant and novel, the manuscript has significant scientific, methodological, and interpretative limitations that must be addressed before consideration for a high-impact medical journal.
Major Concerns
1. The study conclusion that serum Cer16/S1P ratio may serve as a surrogate marker of disease activity in patients with LN. However, this is based on a significant p-value. The absolute difference may not be clinical relevant 0.27 ± 0.09 vs 0.26 ± 0.06. The significant finding is overflated in the interpretation of results and discussion.
2. The study is cross-sectional, yet the discussion suggests causal and longitudinal interpretations (e.g., “predict flares,” “therapeutic response”). Suggest to limit interpretations strictly to associations; avoid causal language. ACknowledge that a cross-sectional design restricts inference.
3. Treatment confounding is inadequately addressed in this study. Glucocorticoid dose, immunosuppressive use may significantly alter sphingolipid metabolism, yet the treatment distributions differ drastically between LN and non-LN groups (e.g., MMF use). There is no further analysis to remove these confounders as only age and sex were included in the multivariable adjustment. May want to consider including treatment variables in regression models, as well as sensitivity analyses, although the small sample size may limit this.
4. ROC analysis interpretation is overstated AUC of 0.739 is moderate at best for a diagnostic biomarker. Confidence intervals are wide due to small sample size. LN vs healthy control ROC curves are not clinically relevant; a more relevant comparison is LN vs non-LN SLE, which is not presented.
5. There are also patients with proteinuria > 0.5g/day in the non-LN group. How did the authors determine that these patients are truly non-LN when there is presence of proteinuria?
6. Lack of biological plausibility connecting Cer16 specifically to LN. The discussion provides general sphingolipid biology, but does not explain why Cer16 specifically (not Cer24 or others) should differentiate LN. The discussion should explore more in depth mechanistic rationale for Cer16 in kidney pathology.
7. Statistical analyses require strengthening. Non-Parametric tests may be more relevant for small sample size and subgroups.
8. Missing description of key methods. How was LN defined (ISN/RPS classification?). No description of fasting status for blood draw as lipidomics is highly sensitive to this.
9. Lack of comparison with existing LN biomarkers. The manuscript does not benchmark Cer16/S1P against traditional markers such as proteinuria, anti-dsDNA, complements. Perhaps the manuscript can explore how Cer16/S1P adds incremental value over existing biomarkers. If existing markers are already enough to differentiate LN patients from HC, why is there a need to explore Cer16/S1P?
While the manuscript contributes interesting data, The most critical issues include the very small LN sample size, inadequate adjustment for treatment confounding, and overinterpretation of results which may not be clinically significant.
Reviewer 3 Report
Comments and Suggestions for Authors
The current manuscript describes the ratio of ceramide (16:0) to sphingosine-1-phosphate as a biomarker for lupus nephritis. Although this has been tackled by other authors, the current manuscript fills in the gap concerning the Korean population and adds up to our understanding of the disease. The work is well-designed and executed, the manuscript is clearly written and comprehensible, whereas the literature is well covered.
Some issues should be addressed, found below
Line 105 how did the authors decide which comparisons to undertake? The ratios will be n x (n-1)/2 diagonally symmetrical matrix. Would a matrix visualization eg a heat map be informative or even better some other statistical method to highlight the statistically significant ratios, e.g. a PCA of loadings would be revealing. As the authors indicated in the results section, there are other ceramides that could serve as plausible targets.
Table 2 please indicate what pvalue1 and pvalue 2 are ie which comparisons do they represent. Also please state what the number in parenthesis are (e.g. confidence intervals?)
Figure 1 Could the authors please state more clearly what are the groups used for the comparisons? I.e. Does the dot plot depict the total SLE vs healthy? The pairwise comparisons with three groups would require three graphs
Also, it would be desirable if the authors could add the uncertainty limits in the form of e.g. confidence intervals to the ROC curve in order to estimate the uncertainty of the graph.
Round 2
Reviewer 1 Report
Comments and Suggestions for Authors
All the points have been adressed and I believe now the Title, Main body and discussion of the paper better reflect the observations of the work.
Author Response
Thank you very much for your thoughtful comments.
We appreciate your careful review, and we are pleased to hear that the revised Title, Main body, and Discussion now better reflect the findings of our work.
Reviewer 2 Report
Comments and Suggestions for Authors
The authors have adequately addressed the concerns. The changes in title as well as discussion section to highlight the exploratory nature of this pilot study now makes this manuscript more coherent. The hypothesis generating nature of this revised manuscript makes it more acceptable. The authors have also made good effort to reanalysed results and addressed the limitations accordingly.
Author Response
Thank you very much for your positive and constructive feedback. We are also grateful for your acknowledgment of the additional analyses and clarification of limitations, which were performed in response to your insightful comments. Your guidance has greatly contributed to strengthening the coherence and overall quality of our manuscript.